🔓 | **Open Peer Review** | Antimicrobial Chemotherapy | Research Article

# Detection of OXA-23 and OXA-58 in *Proteus mirabilis* by immunochromatographic assays

Janina Noster-Schrader,[1,2] Janko Sattler,[3,4,5] Yvonne Stoll,[1] Martina Spille,[1] Andreas Rump,[6] Stephan Göttig,[7] Sören Gatermann,[8] Axel Hamprecht[1,3,4]

**ABSTRACT** *Proteus mirabilis* is increasingly recognized as a carrier of carbapenemases, which are particularly difficult to identify, since most isolates have low carbapenem MICs. Additionally, most phenotypic tests (e.g. CARBA NP, mCIM) have insufficient sensitivity in this species. Furthermore, OXA-23 and OXA-58 are frequent in carbapenemase-producing *Proteus* but not included in most molecular and immunochromatographic assays for carbapenemase detection in *Enterobacterales*. The RESIST ACINETO lateral flow test (CORIS BioConcept) was recently developed for the detection of OXA-23, OXA-58/OXA-40, and NDM in *Acinetobacter* spp. The aim of this study was to investigate whether a combination of the RESIST-5 O.K.N.V.I., developed for *Enterobacterales*, and the RESIST ACINETO could improve the detection of carbapenemases in *P. mirabilis*. Ninety carbapenemase-carrying *P. mirabilis* isolates were analyzed by whole genome sequencing. All isolates were evaluated by the recently published carbapenemase-detection algorithm for *P. mirabilis*, followed by RESIST-5 and RESIST ACINETO assay. The algorithm achieved a sensitivity of 97.8%. In comparison, sensitivity/specificity among targeted carbapenemases was 96.3%/97.2% for RESIST ACINETO and 96.4%/100% for RESIST-5. In combination, a sensitivity of 97.8% was achieved. The sensitivity was excellent for OXA-23 (23/23, 100%) and NDM (18/18, 100%), but lower for OXA-58 (12/13, 92.3%) and OXA-48-like (24/25, 96.0%). The RESIST ACINETO and RESIST-5 performed well in detecting carbapenemases in *P. mirabilis*, particularly for the emerging OXA-23-producing *P. mirabilis*. The assays have the potential to improve carbapenemase detection in the routine diagnostic laboratory for this challenging species.

**IMPORTANCE** OXA-23 and OXA-58 are among the most frequent carbapenemases in *Proteus mirabilis* but are often missed because of the difficult detection. In this article, the performance of immunochromatographic assays for the rapid and easy detection of these enzymes is assessed. The study includes a very large set of molecularly well-characterized isolates. The present work demonstrates that the notoriously challenging detection of carbapenemases in *P. mirabilis* can be achieved by immunochromatographic assays, enabling laboratories outside of academic centers to reliably detect these emerging carbapenemases in a short time. This will help prevent the further spread of carbapenemase-producing *P. mirabilis* in hospitals.

**KEYWORDS** *Proteus* spp., *Acinetobacter* spp., Coris RESIST ACINETO assay, lateral flow assay, OXA-23, OXA-58

**Peer Reviewer** Stefano Mancini, Universitat Zurich, Zurich, Switzerland

Address correspondence to Axel Hamprecht, axel.hamprecht@uol.de.

A.H. has received research support from Coris Biosystems, outside of the presented study. The other authors do not report any conflict of interest.

*Proteus mirabilis*, a member of the family *Morganellaceae*, is a common cause of urinary tract and bloodstream infections (1–3). Several surveillance studies highlighted the increasing frequency of carbapenemase-producing *Proteus* (CPP) isolates globally (4–6). However, numbers are likely underestimated because of their challenging phenotype. Due to the low meropenem and ertapenem MICs in CPP, EUCAST and CLSI

clinical breakpoints typically categorize these isolates as susceptible, making sophisticated algorithms for proper recognition necessary (7). Additionally, most phenotypic confirmation assays currently recommended by CLSI and EUCAST (e.g., mCIM and CARBA NP) perform well in other *Enterobacterales* but fail to reliably detect carbapenemases in *P. mirabilis* (7).

In contrast to the big five carbapenemases—KPC, OXA-48-like, VIM, NDM, and IMP—which are typically present in *Klebsiella pneumoniae* and *Escherichia coli*, *P. mirabilis* is a frequent carrier of OXA-23 and OXA-58 (8). These carbapenemases are usually found in *Acinetobacter* spp. but not in *Enterobacterales*. Therefore, common diagnostic tests for carbapenemase detection in *Enterobacterales* usually do not target OXA-23 and OXA-58, including most commercially available PCR assays such as GeneXpert CARBA-R (9–11).

In recent years, multiplex immunochromatographic assays (ICT) targeting the most prevalent carbapenemases have become available (12). These assays, such as the RESIST-5 O.K.N.V.I assay (CORIS BioConcept), have demonstrated high sensitivity and specificity for the detection of carbapenemases in *Enterobacterales* in several studies (13, 14), including directly from other specimens such as blood cultures (15). However, previous studies have shown a lower performance for carbapenemase detection in *P. mirabilis* compared to other species (13, 16, 17).

The RESIST ACINETO lateral flow test (CORIS BioConcept) has recently been developed for the detection of OXA-23, OXA-40/58, and NDM in *Acinetobacter* spp. We aimed to analyze if the combination of the RESIST-5 O.K.N.V.I for *Enterobacterales* and the RESIST ACINETO assay could improve the detection of carbapenemases in *P. mirabilis*.

## MATERIALS AND METHODS

### Study design and isolate collection

In this study, 90 clinical *P. mirabilis* isolates were included, either isolated between 2015 and 2023 at the university hospitals of Oldenburg, Cologne, and Frankfurt, or obtained from the German National Reference Center for Multidrug-Resistant Gram-Negative Bacteria, Bochum (*n* = 79). Inclusion criteria were positive results for a carbapenemase gene by PCR or a positive result in the modified zinc-supplemented carbapenem inactivation method (mzCIM) assay and/or the recently published carbapenemase detection algorithm (7, 18): isolates exhibiting resistance to ampicillin-sulbactam and/or amoxicillin-clavulanate were further analyzed by agar diffusion testing with temocillin (TEM) and ticarcillin-clavulanate (TCC) disks (Fig. S1). Isolates displaying a TCC zone diameter of ≥20 mm are considered carbapenemase negative, while isolates showing TCC resistance combined with a TEM inhibition zone of <14 mm were classified as carbapenemase positive. This phenotype is most commonly associated with OXA-48-like or OXA-58 enzymes. Isolates with TCC resistance and a TEM inhibition zone of ≥14 mm underwent additional testing using the mzCIM to detect carbapenemase production mediated by other enzymes, e.g., OXA-23, metallo-β-lactamases, or KPC.

### Whole genome sequencing

For DNA isolation, the DNeasy UltraClean Microbial Kit (Qiagen, Hilden, Germany) was used, following the manufacturer's instructions. Isolates were sequenced on the Illumina 6000 platform. Bioinformatic analysis and determination of phylogenetic relatedness were performed as previously described (6, 19). The sequence type (ST) was determined with PubMLST (20) (Fig. S2). Assemblies of some of the strains have been published before on GeneBank under BioProject ID PRJNA915754 (7, 21); assemblies of additional isolates were deposited under BioProject ID PRJNA1201067.

## Analysis of *Proteus* spp. isolates by immunochromatographic lateral flow tests

Isolates were analyzed using RESIST-5 O.K.N.V.I. and RESIST ACINETO lateral flow tests (CORIS BioConcept, Gembloux, Belgium) according to the manufacturer's recommendations. The isolates were cultured on Columbia blood agar plates at 37°C overnight. For the RESIST-5 assay, three colonies were homogenized in 11 drops of lysis buffer. For the RESIST ACINETO assay, a 1 µL loop of bacteria was resuspended in six drops of lysis buffer. For both tests, 100 µL of homogenized bacteria was applied to the respective lateral flow cassette. The results were read after 15 min.

## RESULTS

### Molecular and phylogenetic analysis of carbapenemase-producing *P. mirabilis* isolates

The 90 *P. mirabilis* isolates produced a total of 95 carbapenemases. The majority of isolates ($n = 36$, 40.0%) carried carbapenemase genes of the $bla_{OXA-23/58}$ group ($bla_{OXA23}$: $n = 23$, 25.6%; $bla_{OXA-58}$: $n = 13$, 14.4%) (Table 2; Table S1). Twenty-five isolates (27.8%) harbored carbapenemase genes of the $bla_{OXA-48-like}$ group ($bla_{OXA-48}$: $n = 19$, 21.1%; $bla_{OXA-181}$: $n = 5$, 5.6%; $bla_{OXA-162}$: $n = 1$, 1.1%). Other isolates carried carbapenemase genes of the following groups: $bla_{NDM-1/5}$ ($n = 18$, 20.0%), $bla_{VIM-1/4/75/78}$ ($n = 9$, 10.0%), $bla_{KPC-3}$ ($n = 4$, 4.4%), and $bla_{IMP1}$ (1, 1.1%). Only five isolates harbored more than one carbapenemase gene. The combinations detected were $bla_{VIM-1}/bla_{VIM-4}$, $bla_{VIM1}/bla_{OXA-48}$, $bla_{KPC-3}/bla_{OXA-23}$, $bla_{VIM-4}/bla_{VIM-75}$, and $bla_{NDM1}/bla_{OXA-48}$. Analysis of ST types and phylogeny demonstrated that the isolates had a diverse genetic background. Among OXA-23 producers, the majority ($n = 22$) belonged to the typical ST142 and were highly related, as previously reported from France and Germany (6, 7, 22) (Fig. S2).

### Analysis by the carbapenemase detection algorithm

All 90 isolates were analyzed using the carbapenemase detection algorithm for *Proteus* spp. recently published (7), which includes testing of three antibiotic disks (amoxicillin-clavulanate and/or ampicillin-sulbactam, ticarcillin-clavulanate, and temocillin). Most OXA-48-like (21/25) and OXA-58 (12/13) were directly identified by reduced ticarcillin-clavulanate (<20 mm) and temocillin (<14 mm) inhibition zones, while most other carbapenemases typically had diameters of <20 mm for ticarcillin-clavulanate but ≥14 mm for temocillin, so that the carbapenem inactivation test mzCIM was additionally performed. This was required for 49/85 isolates producing one carbapenemase (57.6%); 20/22 OXA-23 (90.9%), 1/1 IMP-1 (100%), 6/6 VIM-1/78 (100%), 15/17 NDM-1 (88.2%), 3/18 OXA-48 (15.8%), 1/13 OXA-58 (7.7%), 1/4 OXA-181 (25.0%), and 2/3 KPC-3 (66.7%) as well as for 3/5 isolates producing two carbapenemases each (Carb-38, VIM-1, and VIM-4; Carb-62, VIM-4, and VIM-75; Carb-68, KPC-3, and OXA-23).

Of 90 CPPs, 88 isolates (97.8%) were correctly identified by the algorithm as carbapenemase producers, while two isolates gave a false-negative result (Carb-58 and NMD-5, Carb-91 and NDM-1). Therefore, the algorithm achieved a sensitivity of 97.8% (Tables 1 and 2; Table S1).

### Detection of OXA-48, KPC, NDM, VIM, and IMP carbapenemases by O.K.N.V.I. RESIST-5 assay

Of the 90 isolates included in the study, 55 isolates produced carbapenemases which are targeted by the RESIST-5 assay, 4 of them with two detectable carbapenemases each (Table S1). For 53/55 isolates (96.4%), the carbapenemase group could be correctly determined by the ICT, but two isolates showed false-negative results (Carb-87, NDM-1; Carb-25, OXA-48). Thirty-five isolates carried 35 non-targeted carbapenemases (e.g. OXA-23 and OXA-58), and one isolate produced one targeted and one non-targeted carbapenemase (Carb-68, OXA-23, and KPC-3). No false-positive result was recorded. The

RESIST-5 assay thus achieved a sensitivity of 96.4% (53/55) and a specificity of 100% (Tables 1 and 2; Table S1).

## Detection of NDM, OXA-23, and OXA-58 in *P. mirabilis* using the RESIST ACINETO assay

Among the 90 isolates, 54 produced carbapenemases, which are targeted by the RESIST ACINETO assay, whereas 36 isolates produced 39 non-targeted enzymes and 1 isolate produced 1 targeted and 1 non-targeted carbapenemase (Carb-68, OXA-23, and KPC-3) (Table S1). Of these 54 CPPs, 52 isolates (96.3%) were correctly identified as OXA-23 (23/23, 100%), OXA-40/58 (12/13, 92.3%), or NDM producers (17/18, 94.4%). False-negative results were obtained for two isolates (Carb-04, OXA-58; Carb-85, NDM-1). For one VIM-78-producing isolate (Carb-56), the RESIST ACINETO assay yielded a false-positive result as OXA-23. Overall, the RESIST ACINETO assay achieved a sensitivity of 96.3% (52/54) and a specificity of 97.2% (35/36), Tables 1 and 2.

## Combined performance of the RESIST ACINETO and RESIST-5 assay for the detection of carbapenemases in *P. mirabilis*

When both tests were combined for the analysis of the tested isolates, 88/90 isolates were correctly identified as producers of the respective carbapenemases (Table 1; Table S1). False-negative results were obtained for two isolates (Carb-04, OXA-58; Carb-25, OXA-48) and false-positive results for one isolate (Carb-56, VIM-78 detected as OXA-23 positive). The overall sensitivity of the combined assays was 97.8% (88/90, Table 1).

## DISCUSSION

Commercially available assays for detection of carbapenemases target the common enzymes in the most frequent species or orders. The high proportion of *P. mirabilis* isolates carrying the carbapenemases OXA-23 and OXA-58, which are normally known in *Acinetobacter* spp., increases the risk of non-detection and, in the worst case, sub-optimal antibiotic treatment of patients (7).

In this study, two different ICTs and their combination were assessed for detection of carbapenemases in *P. mirabilis*, including the novel RESIST ACINETO assay.

When both tests were combined for the analysis of 90 tested isolates, the assays achieved a sensitivity of 97.8%. When analyzing the isolates that gave false-negative results in one of the two tests, Carb-85 and Carb-87 were both carriers of $bla_{NDM-1}$. While the RESIST-5 assay gave a negative result for Carb-87 and a positive result for Carb-85, the RESIST ACINETO yielded the opposite results for these isolates. This result was reproducible in several repetitions. The carbapenemase detection algorithm showed a clear positive result for both isolates, indicating expression of NDM carbapenemase in both isolates (18). Weak signals for NDM carbapenemase have also been reported when the RESIST ACINETO assay was applied to *Acinetobacter* spp. isolates (23), implying less effective detection of NDM compared to other enzymes. Comparable results for the less sensitive detection of NDM carbapenemases compared to, e.g., OXA-48-like and KPC carbapenemases have also been reported for other *Enterobacterales* using lateral flow tests, especially in *P. mirabilis* (13, 16, 17). The other isolates with false-negative results (Carb-04, OXA-58, and Carb-25, OXA-48) showed positive results in the algorithm but also negative results in the phenotypic mzCIM assay (22 and 24 mm inhibition

**TABLE 1** Comparison of CORIS O.K.N.V.I. RESIST-5 and RESIST ACINETO and the previously published algorithm for identification of carbapenemase positive isolates (calculated relative to isolate numbers)

| | Sensitivity |
|---|---|
| CPP algorithm (7) | 97.8% (88/90) |
| O.K.N.V.I. RESIST-5 | 96.4% (53/55) |
| RESIST ACINETO | 96.3% (52/54) |
| O.K.N.V.I. RESIST-5 + RESIST ACINETO | 97.8% (88/90) |

**TABLE 2** Performance of CORIS O.K.N.V.I. RESIST-5 and RESIST ACINETO and the previously published algorithm for detection of carbapenemases, stratified according to carbapenemase variant

| Enzyme | CPP algorithm (6) | O.K.N.V.I. RESIST-5 | | RESIST ACINETO | |
| --- | --- | --- | --- | --- | --- |
| | True pos. (sensitivity) | True pos. (sensitivity) | False pos. (specificity) | True pos. (sensitivity) | False pos. (specificity) |
| **Ambler class A (*n* = 4)** | **4 (100%)**[b] | **4 (100%)** | **0 (100%)** | **–**[a] | **–** |
| KPC-3 (*n* = 4) | 4 (100%) | 4 (100%) | 0 (100%) | – | – |
| **Ambler class B (*n* = 30)** | **28 (93.3%)** | **29 (96.7%)** | **0 (100%)** | **17 (94.4%)** | **1 (96.8%)** |
| NDM-1 (*n* = 17) | 16 (94.1%) | 16 (94.1%) | 0 (100%) | 16 (94.1%) | 0 (100%) |
| NDM-5 (*n* = 1) | 0 (0%) | 1 (100%) | 0 (100%) | 1 (100%) | 0 (100%) |
| VIM-1 (*n* = 4) | 4 (100%) | 4 (100%) | 0 (100%) | – | – |
| VIM-4 (*n* = 2) | 2 (100%) | 2 (100%) | 0 (100%) | – | |
| VIM-75 (*n* = 1) | 1 (100%) | 1 (100%) | 0 (100%) | – | – |
| VIM-78 (*n* = 4) | 4 (100%) | 4 (100%) | 0 (100%) | – | 1 (80%) |
| IMP (*n* = 1) | 1 (100%) | 1 (100%) | 0 (100%) | – | – |
| **Ambler class D (*n* = 61)** | **61 (100%)** | **24 (96%)** | **0 (100%)** | **35 (97.2%)** | **0 (100%)** |
| OXA-23 (*n* = 23) | 23 (100%) | – | – | 23 (100%) | 0 (100%) |
| OXA-58 (*n* = 13) | 13 (100%) | – | – | 12 (92.3%) | 0 (100%) |
| *OXA-48-like* | | | | | |
| OXA-48 (*n* = 19) | 19 (100%) | 18 (94.7%) | 0 (100%) | – | – |
| OXA-181 (*n* = 5) | 5 (100%) | 5 (100%) | 0 (100%) | – | – |
| OXA-162 (*n* = 1) | 1 (100%) | 1 (100%) | 0 (100%) | – | – |

[a]"–" indicates that the respective test is not applicable for the corresponding carbapenemase.
[b]Bold indicates results according to Ambler class categories.

zone, respectively), indicating low expression of the carbapenemase genes. Thus, the reduced amount of antigen in the sample could potentially make it difficult to detect the respective carbapenemases by lateral flow tests. However, both were positive by whole genome sequencing, by PCR, and in the previously published algorithm for the detection of carbapenemases in *P. mirabilis* (7).

When comparing the overall performance of the RESIST ACINETO assay for the detection of the respective carbapenemases in *Acinetobacter* spp. and *P. mirabilis*, the sensitivity and specificity in *Acinetobacter* isolates were slightly higher in previous studies (99%–100% and 96%–100%, respectively) (23–26). The reasons for the slightly better detection of the investigated carbapenemases in *Acinetobacter* spp. isolates may range from lower expression, less efficient protein synthesis, or structural differences in *Acinetobacter* spp. and *P. mirabilis*.

A recent study analyzed the performance of the NG-Test DectTool OXA-23 assay (NG-Biotech, France) for the detection of OXA-23 in 196 *A. baumannii*- and 8 *P. mirabilis*-positive blood cultures (clinical or spiked samples). The method reached a sensitivity and specificity of 100% but does not allow the detection of OXA-58 (27). Similar to this example, most previous studies on carbapenemase detection methods included only a few or no carbapenemase-producing *P. mirabilis* isolates (28–33). This is likely the reason why the knowledge on CPP is very limited compared to *K. pneumoniae* or *E. coli*. Recently, it has been shown that confirmation assays recommended by EUCAST and CLSI have a weak performance in *P. mirabilis*, with a sensitivity of 30% for CARBA NP and 63% for mCIM (7).

Given the good performance of the novel ICT, the short hands-on time and easy-to-follow protocol RESIST-5 and RESIST ACINETO assay have the potential to improve carbapenemase detection in this challenging species, especially for the emerging OXA-23 carbapenemase. Although the RESIST ACINETO assay was developed for the characterization of *Acinetobacter* isolates, it is also well suited for *Proteus* spp. isolates. Its use is strongly recommended to reduce the false-negative detection of carbapenemases in *Proteus* spp. when methods like the recently published algorithm for the detection of carbapenemases in *Proteus* spp. are not available (7). In this scenario, the RESIST-5 test can be applied immediately on isolates with non-wildtype ertapenem or meropenem

MICs (>0.125 mg/L) and resistance to ampicillin-sulbactam and/or amoxicillin-clavulanate. In case of a negative result, it is recommended to subsequently use the RESIST ACINETO assay. The application of both ICT covers the set of carbapenemases generally found in *Proteus* spp. when rapid identification of the carbapenemase is required but increases the costs per isolate.

There are some limitations in this study. Only *P. mirabilis* isolates from Germany were examined, and only one culture condition was analyzed. The performance of the tests should be evaluated with more isolates in multicenter studies. However, the ICT was challenged with a large collection of well-characterized CPPs, currently representing the largest study on carbapenemases in *Proteus* spp. to the best of our knowledge.

In summary, the combination of both ICTs allows a rapid and simple detection of the whole spectrum of carbapenemases in *Proteus* spp., likely improving infection control and treatment approaches.

## ACKNOWLEDGMENTS

The RESIST-5 and RESIST ACINETO RESIST tests were provided by Coris Biosystems free of charge.

This study was supported by a research grant from the Forschungspool of the Carl von Ossietzky University Oldenburg.

## AUTHOR AFFILIATIONS

[1]Carl von Ossietzky University Oldenburg, Institute of Medical Microbiology and Virology, Oldenburg, Germany

[2]Klinikum Bielefeld, Bielefeld, Germany

[3]German Centre for Infection Research, Partner Site Bonn-Cologne, Cologne, Germany

[4]University Hospital Cologne and Faculty of Medicine, University of Cologne, Institute for Medical Microbiology, Immunology and Hygiene, Cologne, Germany

[5]Department of Machine Learning and Systems Biology, Max Planck Institute of Biochemistry, Martinsried, Germany

[6]Department of Medical Genetics, Carl von Ossietzky University Oldenburg, Oldenburg, Germany

[7]Goethe University Frankfurt, University Hospital, Institute of Medical Microbiology and Infection Control, Frankfurt am Main, Germany

[8]National Reference Laboratory for Multidrug-Resistant Gram-negative Bacteria, Department of Medical Microbiology, Ruhr-University Bochum, Bochum, Germany

## AUTHOR ORCIDs

Janina Noster-Schrader  http://orcid.org/0000-0002-8802-6605
Janko Sattler  http://orcid.org/0000-0003-1415-4169
Stephan Göttig  http://orcid.org/0000-0001-6896-5309
Axel Hamprecht  http://orcid.org/0000-0003-1449-5780

## AUTHOR CONTRIBUTIONS

Janina Noster-Schrader, Formal analysis, Investigation, Project administration, Supervision, Writing – original draft, Writing – review and editing | Janko Sattler, Data curation, Formal analysis, Methodology, Writing – review and editing | Yvonne Stoll, Investigation | Martina Spille, Investigation | Andreas Rump, Investigation, Methodology | Stephan Göttig, Methodology, Resources, Writing – review and editing | Sören Gatermann, Resources, Writing – review and editing | Axel Hamprecht, Conceptualization, Funding acquisition, Methodology, Project administration, Supervision, Writing – review and editing

## DATA AVAILABILITY

Assemblies of sequences are available on GenBank under BioProject IDs PRJNA915754 and PRJNA1201067. Raw data of this study are included in the supplemental material (Table S1).

## ADDITIONAL FILES

The following material is available online.

### Supplemental Material

**Supplemental material (Spectrum01899-25-s0001.pdf).** Fig. S1 and S2; Table S1.

### Open Peer Review

**PEER REVIEW HISTORY (review-history.pdf).** An accounting of the reviewer comments and feedback.

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
