## [Reviewer comments · Microbiology Spectrum]

Microbiology Spectrum

Detection of OXA-23 and OXA-58 in *Proteus mirabilis* by immunochromatographic assays

Janina Noster-Schrader, Janko Sattler, Yvonne Stoll, Martina Spille, Andreas Rump, Stephan Göttig, Soeren Gatermann, and Axel Hamprecht

Corresponding Author(s): Axel Hamprecht, Carl von Ossietzky Universitat Oldenburg

Review Timeline:

Submission Date:	July 13, 2025
Editorial Decision:	November 8, 2025
Revision Received:	January 12, 2026
Accepted:	January 26, 2026

Editor: Haifang Zhang

Reviewer(s): Disclosure of reviewer identity is with reference to reviewer comments included in decision letter(s). The following individuals involved in review of your submission have agreed to reveal their identity: Stefano Mancini (Reviewer #1)

Transaction Report:

DOI: <https://doi.org/10.1128/spectrum.01899-25>

Re: Spectrum01899-25 (Detection of OXA-23 and OXA-58 in *Proteus mirabilis* by immunochromatographic assays)

Dear Prof. Axel Georg Hamprecht:

Thank you for the privilege of reviewing your work. Below you will find my comments, instructions from the Spectrum editorial office, and the reviewer comments.

Revision Guidelines

Sincerely,
Haifang Zhang
Editor
Microbiology Spectrum

Reviewer #1 (Comments for the Author):

Noster et al. evaluated the performance of combined LFIA tests for the detection of carbapenemases in Enterobacterales (RESIST-5) and the *Acinetobacter baumannii* complex (RESIST-ACINETO) for the detection of carbapenemases in *Proteus mirabilis* carbapenemase producers (n = 90). The authors show that, upon suspicion of carbapenemase production, reliable detection can be achieved using the proposed algorithm. The work is relevant and timely, as these isolates are not yet

widespread/reported globally, and the study provides valuable guidance on how to handle them.

Minor comments:

- Since all clinical isolates were analyzed by WGS, a figure showing clonal relatedness would help illustrate genetic variability. Such a figure should be added and discussed in the text.
- OXA-40/58 is sometimes written as OXA-58/40; please ensure consistency throughout the manuscript.
- Line 167: In "In case of ticarcillin...", remove the word 'an'.
- Line 113: Rephrase as: "In this study, 90 clinical *P. mirabilis* isolates were included, either isolated..."
- Line 68: Write *Proteus mirabilis* in italics.
- Line 80: Replace 'indicated' with 'highlighted'.
- Table 1: In the header, it currently reads Sensitivity (True pos.), while in the corresponding columns, the true positives are listed first, followed by sensitivity in parentheses. Please correct the table header to make it consistent with the data (e.g. True positives (Sensitivity)).
- Table S1: Indicate the sequence types (ST) of the clinical isolates.
- In the Discussion, please mention when and how the lateral flow immunoassays should be performed and integrated into the diagnostic algorithm. Specify which test should be performed first, and based on the results, if and when a second test should follow.
- Please submit the genome sequences to a public database (such as ENA: <https://www.ebi.ac.uk/ena/browser>) to ensure they are publicly accessible.

Dear reviewers,

We would like to thank you very much for your helpful comments on our manuscript entitled "Detection of OXA-23 and OXA-58 in *Proteus mirabilis* by immunochromatographic assays." We have incorporated your suggestions into the new version of the manuscript. For clarity, we would like to address each point individually:

"Since all clinical isolates were analyzed by WGS, a figure showing clonal relatedness would help illustrate genetic variability. Such a figure should be added and discussed in the text."

Thank you for your comment. We have added the phylogenetic analysis and the corresponding tree in the appendix (Supplementary Figure 2). Corresponding methods and results can be found in lines 120-127 and 146-149.

"OXA-40/58 is sometimes written as OXA-58/40; please ensure consistency throughout the manuscript."

Thank you for pointing out this error. We have corrected the names in lines 95 and 192.

"Line 167: In "In case of ticarcillin...", remove the word 'an'."

The sentence was re-written in lines 157.

"Line 113: Rephrase as: "In this study, 90 clinical *P. mirabilis* isolates were included, either isolated...""

Thank you for the comment, we corrected the sentence in line 100.

"Line 68: Write *Proteus mirabilis* in italics."

Thank you for pointing this out. We have checked the formatting throughout the manuscript. We could not find any missing italics. We only omitted italics for clarity's purposes when the species was in a chapter heading that was written in italics.

"Line 80: Replace 'indicated' with 'highlighted'."

Thank you for this comment. We changed the sentence in lines 70.

"Table 1: In the header, it currently reads Sensitivity (True pos.), while in the corresponding columns, the true positives are listed first, followed by sensitivity in parentheses. Please correct the table header to make it consistent with the data (e.g. True positives (Sensitivity))."

Thank you for your helpful feedback. We have modified the table accordingly (lines 432-434).

“Table S1: Indicate the sequence types (ST) of the clinical isolates.”

Thank you very much, we have carried out the relevant analysis and listed the STs in the table. Additionally, you can find the STs in Supplementary Figure 2.

„In the Discussion, please mention when and how the lateral flow immunoassays should be performed and integrated into the diagnostic algorithm. Specify which test should be performed first, and based on the results, if and when a second test should follow.”

Thank you for your helpful comment. We have added the relevant information in lines 254-262.

“Please submit the genome sequences to a public database (such as ENA: <https://www.ebi.ac.uk/ena/browser>) to ensure they are publicly accessible.”

The data is available in GeneBank; the corresponding information can be found in lines 122-125.

Sequencing was repeated for three isolates as part of the revision (Carb-55, Carb-62, Carb-65). The results had no impact on the overall results of the publication. In the repeated analysis, Carb-62 showed VIM-4 and VIM-75 (both VIM-1 like) instead of VIM-1. Additionally, carbapenemases in four isolates previously identified as VIM-4 were changed to VIM-78 (Carb-50, Carb-56, Carb-61, Carb-78). VIM-78 was previously not included in the resistance gene database (and still missing in RESFINDER). These changes required adaptations of numbers, however, those small changes did not change the results of the publication.

We would like to thank you once again for your comments, which have improved our manuscript in various respects.

Re: Spectrum01899-25R1 (Detection of OXA-23 and OXA-58 in *Proteus mirabilis* by immunochromatographic assays)

Dear Prof. Axel Georg Hamprecht:

Your manuscript has been accepted, and I am forwarding it to the ASM production staff for publication. Your paper will first be checked to make sure all elements meet the technical requirements. ASM staff will contact you if anything needs to be revised before copyediting and production can begin. Otherwise, you will be notified when your proofs are ready to be viewed.

Sincerely,
Haifang Zhang
Editor
Microbiology Spectrum

Reviewer #1 (Comments for the Author):

The authors have addressed all the reviewers' comments, revised the manuscript accordingly, and updated the figures where appropriate.